# Characterization and Antioxidant Activity of Exopolysaccharides Produced by *Lysobacter soyae* sp. nov Isolated from the Root of *Glycine max* L.

**DOI:** 10.3390/microorganisms11081900

**Published:** 2023-07-27

**Authors:** Inhyup Kim, Geeta Chhetri, Yoonseop So, Sunho Park, Yonghee Jung, Haejin Woo, Taegun Seo

**Affiliations:** Department of Life Science, Dongguk University-Seoul, Goyang 10326, Republic of Korea; inhyup91@gmail.com (I.K.); geetachhetri123@yahoo.com (G.C.); sts5552@naver.com (Y.S.); tjsgh1837@gmail.com (S.P.); joh2395@naver.com (Y.J.); woohj999@naver.com (H.W.)

**Keywords:** rhizosphere, bacterial exopolysaccharide, *Lysobacter*, *Glycine max* L.

## Abstract

Microbial exopolysaccharides (EPSs) have attracted attention from several fields due to their high industrial applicability. In the present study, rhizosphere strain CJ11^T^ was isolated from the root of *Glycine max* L. in Goyang-si, Republic of Korea, and a novel exopolysaccharide was purified from the *Lysobacter* sp. CJ11^T^ fermentation broth. The exopolysaccharide’s average molecular weight was 0.93 × 10^5^ Da. Its monosaccharide composition included 72.2% mannose, 17.2% glucose, 7.8% galactose, and 2.8% arabinose. Fourier-transform infrared spectroscopy identified the exopolysaccharide carbohydrate polymer functional groups, and the structural properties were investigated using nuclear magnetic resonance. In addition, a microstructure of lyophilized EPS was determined by scanning electron microscopy. Using thermogravimetric analysis, the degradation of the exopolysaccharide produced by strain CJ11^T^ was determined to be 210 °C. The exopolysaccharide at a concentration of 4 mg/mL exhibited 2,2-diphenyl-1-picrylhydrazyl free-radical-scavenging activity of 73.47%. Phylogenetic analysis based on the 16S rRNA gene sequencing results revealed that strain CJ11^T^ was a novel isolate for which the name *Lysobacter soyae* sp. nov is proposed.

## 1. Introduction

Soil is a repository of various microorganisms, and soil microorganisms inhabiting it are very important to humans because they also affect air quality [1]. Exopolysaccharides (EPSs) are biological polymers secreted by microorganisms to cope with harsh environmental conditions such as antibiotics, pH, osmotic stress, and host immune defenses [2,3,4]. Naturally occurring biopolymers are produced by living organisms and the like [5]. Polysaccharides are considered natural polymers because they are complex polymers composed of monosaccharide chains linked by glycosidic bonds and are considered nontoxic, with good biocompatibility [6,7]. EPSs produced by microorganisms are polymer materials widely used in various industries [8,9,10]. The biopolymers produced by bacteria have different chemical properties, and some of these biopolymers have physiologically active functions [11,12,13]. Bacterial EPSs perform anti-inflammatory, immunomodulatory, and antioxidant functions [14]. Moreover, bacterial EPS production is one way to survive changes in the microenvironment, which prevents plants from drying out due to their high water-holding capacity and improves survival by avoiding the drying of bacterial cells [15].

To withstand soil drying, rhizosphere microbes employ a variety of strategies to maintain high water content using exopolysaccharides that are characterized as hygroscopic. This will help to keep the plant growing and prevent the roots from drying [16,17]. Additionally, EPS not only protects plants from drought stress, but also helps bacteria to attach to plant roots [18]. Reactive oxygen species (ROS) are byproducts of oxygen metabolism and play an important role in maintaining homeostasis and redox balance [19]. However, since excessive ROS can cause damage to the body, including cancer, diabetes, aging, and chronic diseases, it is important to reduce the negative aspects. Finding natural antioxidants rather than chemical ones is extremely important [20,21]. Using DPPH free radicals, it was confirmed that EPS produced by strain CJ11 had free-radical-scavenging activity [22].

The genus *Lysobacter*, belonging to the family *Lysobacteraceae* within the class *Lysobacterales*, was first proposed by Christensen and Cook as a nonfruiting, gliding bacterium [23], and most of the species in this genus are rhizosphere, freshwater, and soil-dwelling organisms [23,24]. *Lysobacter* species are Gram-negative, aerobic, nonfruiting, gliding bacteria. The *Lysobacter* strain CJ11^T^ isolated in this study was isolated from the roots of *Glycine max* L. and showed typical *Lysobacter* species characteristics. The genus *Lysobacter* consists of more than 70 species, which have potential in waste degradation and biotechnology because they can produce enzymes such as proteases, chitinases, and lipases [25,26,27,28]. In addition, some strains have been found to inhibit biofilm formation or produce antioxidants, biocontrol agents, and antibiotics, and they are expected to have potential in various industries [13,29,30]. Antibacterial compounds produced by *Lysobacter* include maltophilin, dihydromaltophilin, lysobactin, tripopeptin, phenazine, and lactivicin [30]. The potential of the *Lysobacter* species for protection against plant pathogenicity has been asserted in some studies [31,32]. In this study, the EPS produced by the *Lysobacter* species isolated from the rhizosphere of beans in the Republic of Korea was characterized by Fourier-transform infrared (FTIR) analysis, bio-liquid chromatography (Bio-LC), gel permeation chromatography (GPC), and nuclear magnetic resonance spectroscopy (NMR). Furthermore, the EPS showed DPPH-scavenging ability, thus displaying potential antioxidant capability. Moreover, on the basis of phylogenetic and polyphasic analyses, the isolated *Lysobacter* species is proposed as a novel species in the genus *Lysobacter*. Among the strains isolated from the rhizome (soybean field) in Ilsan, Korea, strains that were confirmed to produce EPS and exhibited DPPH free-radical-scavenging activity were selected. Accordingly, CJ11^T^, a rhizosphere strain, was proposed as a novel species in the genus *Lysobacter* through phylogenetic and polyphasic analysis. The study also provides theoretical support for industrial applications of the EPS produced by strain CJ11^T^.

## 2. Materials and Methods

### 2.1. Isolation, 16S rRNA Gene Analysis, Physiology, and Chemotaxonomy Characterization

A novel bacterial strain, designated CJ11^T^, was isolated from the root of *Glycine max* L. in Goyang-si, Republic of Korea (Appendix A; 37°40′32.0″ N, 126°48′19.7″ E). To purify this novel strain, a standard dilution method was employed, as described previously, using Reasoner’s 2A agar (R2A; MB cell, Seoul, Republic of Korea) plates [33]. One hundred microliter aliquots of the aforementioned sample suspensions were spread onto R2A plates supplemented with 1% glucose (*w*/*v*) and subsequently incubated at 30 °C for 3 days. The purification procedure was repeated four times. *Lysobacter tolerans* UM1^T^ and *Lysobacter silvestris* AM20-91^T^ were obtained from The Leibniz Institute DSMZ German Collection of Microorganisms and Cell Cultures GmbH (DSMZ; Braunschweig-Süd, Germany). The selected CJ11^T^ strain and reference strains were stored in 25% glycerol (*w*/*v*) at −80 °C. All strains used in taxonomic experiments were cultured on R2A agar plates or in R2A broth at 30 °C for 72 h. The isolated CJ11^T^ strain was deposited in the Korean Agricultural Culture Collection (KACC, Jeonju, Republic of Korea) and the National Institute of Technology and Evaluation (NITE) Biological Resource Center (NBRC, Shibuya-ku, Japan) (CJ11^T^ = KACC 21716^T^ = NBRC 114478^T^).

The 16S rRNA gene sequence of strain CJ11^T^ was amplified using universal bacterial primer sets 27F, 518F, 805R, and 1492R, and sequencing was performed by SolGent Co., Ltd. (Daejeon, Republic of Korea) [34]. The nearly complete sequence of the 16S rRNA genes (1469 bp) was assembled using the SeqMan 2 software (DNASTAR Inc., Madison, WI, USA), and full-length 16S rRNA gene sequences (1547 bp) were extracted from the genome using Basic Rapid Ribosomal RNA Predictor (Barrnap) (0.9-dev) (https://github.com/tseemaan/barrnap (accessed on 13 September 2022)); both yielded identical results. In order to compare the complete sequence of 16S rRNA genes with those of the other taxa, the National Center for Biotechnology Information (NCBI) Basic Local Alignment Search Tool (BLAST) and the EzBioCloud.net (https://www.ezbiocloud.net/ (accessed on 4 August 2022)) databases were searched [35,36]. Multiple sequences were aligned using the Molecular Evolutionary Genetics Analysis (MEGA) 11 software and analyzed using CLUSTAL W [37,38]. Phylogenetic trees were constructed according to the neighbor-joining (NJ) and maximum-likelihood (ML) methods employing the Kimura two-parameter model [39]. The min-mini heuristic algorithm was applied to the maximum-parsimony (MP) method to compare the phylogenetic trees that were constructed using the neighbor-joining method [40]. MEGA 11 software was used to not only construct neighbor-joining trees to estimate the confidence of tree topologies, but also construct phylogenetic trees using bootstrap analyses with 1000 replications [41].

The Gram staining reaction for the strain was performed using a previously described method [42]. To identify the morphology of strain CJ11^T^, cells that were grown in R2A agar at 30 °C for 3 days were negatively stained using 3% uranyl acetate and observed under a transmission electron microscope (TEM; Libra 120; Zeiss, Oberkochen, Germany). The growth of strain CJ11^T^ was examined on R2A, marine agar (MA; MB cell), nutrient agar (NA; Difco, Franklin Lakes, NJ, USA), Luria–Bertani agar (LB; Difco), and tryptic soy agar (TSA; Difco) media at 30 °C for 10 days to identify the optimal medium. The growth of strain CJ11^T^ was assessed at temperatures of 2, 4, 10, 15, 25, 30, 35, 37, 40, and 42 °C on R2A and MA for 10 days. In addition, NaCl tolerance was tested by culturing the strains in R2A broth containing various NaCl concentrations ranging from 0% to 8% (with intervals of 1%) for 10 days. The optimal pH for growth was determined by culturing the cells in R2A broth made using four different buffers with pH levels ranging from 5.0 to 11.0 (with 1 unit pH intervals) and subsequently incubating these cultures at 30 °C for 10 days. The pH level was modulated using the following filter sterilized buffers at a final concentration of 50 mM: acetate buffer (pH 5.0), phosphate buffer (pH 6.0–8.0), Tris buffer (pH 9.0–10.0), and Na_2_HPO_4_/NaOH buffer (pH 11.0). A GasPak jar (BBL, Cockeysville, MD, USA) was used to assess bacterial growth under anaerobic conditions on R2A plates at 30 °C for 10 days. An oxygen absorber strip (Mitsubishi Gas Chemical, Tokyo, Japan) was used and continuously monitored to remove oxygen in the anaerobic chamber. Catalase activity was observed by detecting oxygen bubble production using a 3% (*v*/*v*) aqueous hydrogen peroxide solution, and oxidase activity was observed through the oxidation of 1% (*w*/*v*) tetramethyl-*p*-phenylenediamine (BioMérieux, Durham, NC, USA). Motility was observed by employing a 0.4% agar stabbing technique (tube method), and gliding motility was tested using the hanging-drop technique [43]. Hydrolysis of DNA (DNase agar; MB cell), CM-cellulose (2%; Duksan, Seoul, Republic of Korea), and casein (2% skim milk powder; Biopure, Seoul, Republic of Korea) was tested on R2A, as described previously [44]. The presence of flexirubin-type pigments was investigated using a 20% potassium hydroxide (KOH) solution (*w*/*v*) [45]. Biochemical and enzymatic tests were performed using the API 20NE kit according to the manufacturer’s instructions (BioMérieux).

CJ11^T^ cells grown on R2A plates at 30 °C for 3 days were used to analyze quinone and polar lipid contents. The polar lipid extracts were separated via two-dimensional thin layer chromatography (TLC) by employing two different development solvents, with a chloroform–methanol–water ratio of 65:25:4 (*v*/*v*/*v*) and a chloroform–acetic acid–methanol–water ratio of 80:15:12:4 (*v*/*v*/*v*/*v*). The results were visualized by spraying with Zinzadze’s reagent (molybdenum blue spray reagent, 1.3%; MilliporeSigma, St. Louis, MO, USA) to detect phospholipids, molybdophosphoric acid (phosphomolybdic acid reagent, 5% *v*/*v* solution in ethanol; Sigma, Kawasaki, Kanagawa, Japan) to detect total lipids, *α*-naphthol reagent to detect glycolipids, and ninhydrin reagent (0.2% solution; Sigma) to detect amino lipids [46]. Isoprenoid quinones were extracted using chloroform and methanol at a ratio of 2:1 (*v*/*v*) and were analyzed using high-performance lipid chromatography by following a previously published method [47,48].

### 2.2. Genome Features

Genomic DNA was extracted using the Universal Genomic DNA Extraction Kit (Takara Bio, San Jose, CA, USA) following the manufacturer’s protocol. The draft genome sequencing of strain CJ11^T^ libraries was performed using the Illumina HiSeq × platform (Illumina, San Diego, CA, USA). The reads were assembled using the SPAdes ver. 3.14.1 de novo assembler [49]. The bioinformatics tool CheckM was used to analyze the completeness and contamination of strain CJ11^T^ [50]. A phylogenomic tree was constructed using an up-to-date bacterial core gene set (UBCG) and whole-genome sequences of the closely related genera were obtained from the EzBioCloud Whole-Genome database [51]. The average nucleotide identity (ANI) values were achieved using the e-service of EzBioCloud [52]. The ANI between the novel strain and its close relatives was calculated using KBase wrapper for Fast ANI (https://github.com/ParBLiSS/FastANI (26 April 2023)) [53,54]. The estimated digital DNA–DNA hybridization values were analyzed using the Genome-to-Genome Distance Calculator 3.0 (GGDC; http://ggdc.dsmz.de (accessed on 25 April 2023)) [55]. The average amino-acid sequence identity was analyzed and calculated using EzAAI v1.2.2. [56], and the DNA G + C content of strain CJ11^T^ was calculated from the draft genome. Genes involved in secondary metabolism were predicted using antibiotics and the secondary metabolite analysis shell (antiSMASH) 6.0 [57]. The draft genome was annotated using the Rapid Annotation using Subsystems Technology (RAST) [58,59]. Additionally, the draft genomes were analyzed with PROKKA (v1.14.6), and the location of the tRNA genes, protein coding sequences, and rRNA genes was confirmed [60]. Functional annotation was conducted in the eggNOG 6.0 database of strain CJ11^T^ and the reference strains [61]. The OrthoVenn3 web server was utilized to analyze the comparison and annotation of orthologous gene clusters among strain CJ11^T^, *L. tolerans* UM1^T^, and *L. silvestris* AM20-91^T^ genomes [62]. The genome of strain CJ11^T^ was constructed as a circular functional genome map using the Circular Genome Viewer (CGView) server [63]. Additionally, the metabolic predictions of CJ11^T^ and the related phylogenetic species were made using Distilled and Refined Annotation of Metabolism (DRAM) [64].

### 2.3. EPS Kinetics and Bacterial Growth

Strain CJ11^T^ (OD_600_ = 0.6) was inoculated into fresh sterile R2A broth supplemented with 1% glucose, galactose, and mannose, respectively, incubated at 30 °C at 150 rpm for 7 days and monitored. Bacterial growth and EPS production were observed every 24 h to determine when the batch fermentation was ready for harvesting [65]. Briefly, 2 mL of the culture was collected every 24 h, aliquoted into 96 wells, and measured in a microplate at 600 nm. The EPS yield was determined by centrifuging the culture medium, adding three times the volume of 100% ethanol to 100 mL of the cell-free supernatant, collecting the precipitated EPS overnight, and measuring the weight after freeze-drying.

### 2.4. Extraction and Purification of EPS

To produce EPS, 1 L of fresh medium supplemented with 1% glucose in a 3 L Erlenmeyer flask was inoculated with 1% of the inoculum that was cultured when measuring bacterial growth. Shaking of the culture was performed in an incubator at 30 °C and 150 rpm for 5 days. After centrifugation (8000× *g*, 20 min) to obtain a cell-free supernatant, 14% trichloroacetic acid (TCA) was added thereto, and the solution was cultured with shaking at 90 rpm at room temperature for 30 min. Thereafter, centrifugation was performed again under the same conditions to remove the denatured protein. Ice-cold absolute ethanol three times the volume of the upper layer was added, and the EPSs were precipitated overnight in a refrigerator at 4 °C. After separating the precipitated EPSs and completely removing the residual ethanol, the EPSs were placed in a dialysis membrane and dialyzed (10 K MWCO, SnakeSkin Dialysis Tubing, Thermo Scientific, Branchburg, NJ, USA) with ultrapure water (UPW) for 72 h, and the UPW was replaced every 24 h. Confirmation of protein removal was performed by measuring absorbance at 595 nm to confirm that protein was removed in the EPS by Bradford assay [66]. The purified EPS was then weighed on a balance after lyophilization to determine the yield and stored in a −80 °C deep freezer for further experiments.

### 2.5. Chemical Analysis of EPS

The total carbohydrate content of the EPS was calorimetrically measured using the phenol–sulfuric acid method by drawing a standard curve based on _D_-glucose according to a previously described method [67]. The total carbohydrate content of the EPS was measured spectrophotometrically at an absorbance of 490 nm. Lowry analysis was used to determine the protein content of the EPS, and bovine serum albumin (BSA) was used as the calibration standard [68]. Standard curves were generated using various concentrations of BSA (0 to 2 mg/mL) according to the manufacturer’s instructions. (Pierce™ BCA Protein Assay Kit; Thermo Fisher Scientific, Waltham, MA, USA). Samples were measured at an absorbance of 562 nm using a spectrophotometer (Multiskan GO; Thermo Fisher Scientific, Waltham, MA, USA).

### 2.6. Monosaccharide Composition of EPS

Monosaccharide analysis was performed by bioliquid chromatography (Bio-LC). EPS samples were hydrolyzed to perform monosaccharide compositional analysis. A 2 mg sample of EPS was hydrolyzed with 2 mL of 2 M trifluoroacetic acid. The monosaccharide composition was determined by Bio-LC using a Dionex™ Carbopac™ PA-20 anion-exchange chromatography column (ICS-5000PC; Thermo Dionex, Rommerskirchen, Germany). Peaks were identified using the following standards: mannose, arabinose, glucose, galactose, and rhamnose.

### 2.7. FE-SEM, TEM, and FTIR Analysis

Field-emission scanning electron microscopy (FE-SEM) was used to visualize, observe, and analyze the surface morphology and microstructure of the EPS. Three milligrams of freeze-dried EPSs were attached to carbon tape, mounted on a stub, and coated with gold. EPS observations with FE-SEM were performed at an accelerating voltage of 15 kV. Images of the FE-SEM were observed at 3000×, 10,000×, and 30,000× magnifications. To identify the morphology of strain CJ11^T^, cells that were grown in R2A agar at 30 °C for 3 days were negatively stained using 3% uranyl acetate and observed under a TEM (Libra 120; Zeiss). Functional group identification was evaluated by the FTIR attenuated total reflection (ATR) spectra of the EPS samples. A total of 32 background scans with a resolution of 3 were used in the diamond crystal ATR method. Spectra were acquired from 4000 to 400 cm^−1^ on a Perkin Elmer spectrophotometer.

### 2.8. Mw Determination of EPS

The average molecular weight (Mw) of EPS was assessed by gel permeation chromatography (GPC; HLC-8420; Tosoh, Tokyo, Japan). The TSKgel G2500PW_XL_ column was used, and the Mw of the EPS was estimated using the refractive index (RI) detector. The EPS sample (3 mg/mL; 50 µL) was prepared and eluted with 0.1 M NaNO3 at 40 °C at a flow rate of 1 mL/min. Using the EcoSEC Elite HLC-8420 GPC (Tosoh Biosciences, San Francisco, CA, USA), the Mw of the EPS samples was calculated on the basis of the peak time. Pullulan, a standard with known peak molecular weights (180–642,000 Da; Sigma), was used for calibration. A calibration curve was used to determine the mean average Mw of the EPS.

### 2.9. X-Ray Diffraction (XRD) and Thermogravimetric (TGA) of EPS

After the EPS samples were ground to a fine powder and mounted on a quartz substrate, Ckα X-rays were generated to continuously record intensity peaks using a scintillation counter detector. XRD analysis was performed in the range of 5 to 80 °C (Ultima; Rigaku, Tokyo, Japan). TGA was performed using a Pyris TGA N-1000 model. Ten milligrams of the EPS samples were heated from 25 to 800 °C at a rate of 10 °C/min under nitrogen airflow. The XRD and TGA analyses were performed to evaluate the physical properties of the EPS using the lyophilized EPS powder.

### 2.10. ^1^H- and ^13^C-NMR Analysis of EPS

EPSs produced by *Lysobacter soyae* CJ11^T^ were stored in a lyophilized state at −80 °C for several days. ^1^H- and ^13^C-NMR spectra for the EPS were obtained by dissolving approximately 25 mg of sample in 0.7 mL of deuterium oxide (D_2_O, 99.9%) in an NMR tube (5 mm diameter), and ^1^H- and ^13^C-NMR spectra were taken at 27 °C. Chemical shifts were expressed in parts per million (ppm) on the Bruker 500 MHz FT-NMR spectrometer.

### 2.11. DPPH Free-Radical-Scavenging Activity

The DPPH radical-scavenging capacity of the EPS produced by *Lysobacter soyae* CJ11^T^ was determined according to a previously reported method with minor modifications [69]. Briefly, 100 µL of DPPH solution (0.2 mM) was mixed with 100 µL of the EPS sample solution at various concentrations (0, 0.5, 1, 2, and 4 mg). After incubation at room temperature for 30 min, 200 µL was transferred to a 96-well microplate, and the absorbance was measured in a microplate reader (517 nm). The antioxidant experiment was conducted in triplicate using ascorbic acid as a positive control and deionized water as a negative control. DPPH free-radical-scavenging activity was calculated using the following equation:Scavenging activity of EPS (%) = [1 − (A_sample_ − A_blank_)/A_control_] × 100%,(1)
where A_sample_ is the absorbance of the DPPH solution mixed with the EPS solution, A_blank_ is the absorbance of the DPPH solution, and A_control_ is the absorbance of the control.

## 3. Results and Discussions

### 3.1. Phylogenetic Analysis, Physiology, and Morphological Characteristics

Strain CJ11^T^ has a single copy of the 1574 bp 16S rRNA gene. As a result of the EzBioCloud search based on the 16S rRNA gene sequence, the CJ11^T^ strain was found to be closely related to *L*. *tolerans* UM1^T^ (98.2%) and *L*. *silvestris* AM20-91^T^ (97.8%). In the NJ, ML, and MP phylogenetic trees based on 16S rRNA gene sequences, strain CJ11^T^ formed consistent clusters with two species of genus *Lysobacter* (phylum, *Pseudomonadota*; class, *Gammaproteobacteria*; order, *Lysobacteriales*). The NJ phylogenetic tree method revealed similar topologies, wherein strain CJ11^T^ formed a cluster with *L*. *tolerans* UM1^T^ and *L*. *silvestris* AM20-91^T^ (Figure 1). This relationship was also observed in trees reconstructed using the MP and ML phylogenetic trees with similar topologies (not shown). These results suggest that strain CJ11^T^ belongs to the family *Lysobacteraceae* and is a novel species in the genus *Lysobacter*. On the basis of 16S rRNA gene sequence analysis, *L. tolerans* UM1^T^ and *L. silvestris* AM20-91^T^ were selected for further phenotypic and chemotaxonomic comparisons.

Cells of strain CJ11^T^ were observed to be Gram-negative, strictly aerobic, nonmotile, non-spore-forming, and rod-shaped, which revealed the absence of flagella. Colonies of strain CJ11^T^ were circular, yellow, convex, and smooth on R2A and TSA agar. Strain CJ11^T^ grew well on TSA, R2A, and NA agar in descending order, and only slightly on LB agar; however, the cells did not grow on MA agar. Growth occurred at pH 5.0–11.0 (optimum, pH 7.0–8.0), with 0–2% NaCl (optimum, 0%; *w*/*v*), and at 15–37 °C (optimum, 30 °C). Strain CJ11^T^ also tested positive for catalase and negative for oxidase activity. It was also negative for the hydrolysis of CM-cellulose, casein, chitin, Tween-80, and DNase; however, its hydrolysis of Tween-20 was positive. Flexirubin-type pigments were absent. Biochemical and phenotypic characteristics of strain CJ11^T^ were compared with those of the reference strains and are presented in Appendix A.

The total polar lipid profile of strain CJ11^T^ was found to contain phosphatidylethanolamine (PE), diphosphatidylglycerol (DPG), phosphatidylglycerol (PG), and two unidentified phospholipids (PL1-2) (Appendix A). Although the major polar lipid profile of strain CJ11^T^ was similar to that of the phylogenetically related *Lysobacter* species, the presence of minor polar lipids differentiates it from other closely related species [70,71]. Ubiquinone Q-8 was identified as the respiratory quinone.

### 3.2. Genome Features of Strain CJ11^T^

The genome sequence size for strain CJ11^T^ was determined to be 2,135,237 bp with one contig, an N50 contig of 2,135,137 bp, and a DNA G + C content of 59.2 mol.% (Appendix A). The genome of strain CJ11^T^ encoded 2069 genes in total, containing 2007 protein coding genes (Appendix A). CheckM revealed that the completeness of the CJ11^T^ strain genome was 97.8% with a contamination level of 0.17%. On the basis of the constructed 92 core genes using the UBCG method, the phylogenomic tree was constructed to show the genomic evolutionary distance of the species in the family *Lysobacteraceae* (Appendix A). The tree shows that the closest phylogenetic neighbors were of *L*. *tolerans* UM1^T^ and *L*. *silvestris* AM20-91^T^, similar to the results of the NJ, ML, and MP phylogenetic trees based on the 16S rRNA genes and genetic relatedness. ANI values were calculated with FastANI using orthogonal mapping, and then the genomic conservation of strain CJ11^T^ and the two phylogenetically close strains were visualized (Figure 2). Reciprocal mappings between the variant CJ11^T^ and the reference genome are shown as red lines, indicating evolutionarily conserved regions. The ANI values between strain CJ11^T^ and *L. tolerans* UM1^T^ and *L. silvestris* AM20-91^T^ were 71.6% and 72.0%, respectively, with respective in silico DNA–DNA hybridization values of 19.5% (17.3–21.9%) and 18.5% (16.4–20.9%). These values are considerably below the ANI threshold of 95%, which facilitated the discrimination of the bacterial species [72]. ANI values between strain CJ11^T^ and *L. tolerans* UM1^T^ and *L. silvestris* AM20-91^T^ were 65.4% and 65.9%, respectively. The ANI values between strain CJ11^T^ and other species of *Lysobacteraceae* are shown in Appendix A. The antiSMASH server revealed one secondary metabolite biosynthetic gene cluster for aryl polyene (located from 317,699 to 359,837). The aryl polyene biosynthetic clusters involved in the production of flexirubin pigments are structurally similar to carotenoid pigments and are widespread in bacteria [73]. Strain CJ11^T^ was experimentally found to have the flexirubin pigmentation. A total of 26 cell-wall- and capsule-associated proteins in the genome of CJ11^T^ were predicted. Among them, 11 proteins belonged to the capsular and extracellular polysaccharide part, one protein belonged to the Gram-negative cell-wall component, and 14 unclassified proteins were predicted. Under the capsular and extracellular polysaccharide subcategories were dTDP-rhamnose synthesis (five) and rhamnose-containing glycans (six). According to the results of the NCBI Prokaryotic Genome Annotation Pipeline (PGAP), strain CJ11^T^ has a gene that putatively encodes exopolysaccharide biosynthesis protein (824,161 to 824,799; length 212 bp) and pyruvate glycosyltransferase EpsE. It is known that the pyruvate glycosyltransferase EpsE is required for the initial steps of EPS biosynthesis [74]. Upon the clusters of orthologous groups (COG) classification of strain CJ11^T^, a total of 1965 genes were assigned to 21 functional categories. The eight major parts of the COG categories were as follows: S (function unknown; 23.8%), K (transcription, ribosomal structure, and biogenesis; 7.9%), M (cell wall/membrane/envelope biogenesis; 7.3%), E (amino-acid transport and metabolism; 6.5%), C (energy production and conversion; 6.0%), L (replication, recombination, and repair; 5.9%), O (post-translational modification, protein turnover, and chaperones; 5.6%), and T (signal transduction mechanisms; 5.1%). The overall comparative analysis of strain CJ11^T^ and its phylogenetically related neighbors are shown in Appendix A. A total of 1467 orthologous genes were shared among all three compared species (strain CJ11^T^, *L. tolerans* UM1^T^, and *L. silvestris* AM20-91^T^), of which 130 orthologous genes were shared only between strains CJ11^T^ and *L. silvestris* AM20-91^T^, and 86 orthologous genes were shared between strains CJ11^T^ and *L. tolerans* UM1^T^ (Figure 3). To compare the metabolic abilities between CJ11^T^ and two phylogenetically related *Lysobacter* species, the Refined Annotation of Metabolism function of the Distilled and KBase platforms was used. The DRAM tool provided a metabolic profile for each genome (Figure 4). The metagenome-assembled genome (MAG) of strain CJ11^T^ showed that its genes were involved in glycolysis (Embden–Meyerhof pathway), the pentose phosphate pathway (pentose phosphate cycle), reductive pentose phosphate cycle (Calvin cycle), reductive citrate cycle (Arnon–Buchanan cycle), dicarboxylate–hydroxybutyrate cycle, and reductive acetyl-CoA pathway (Wood–Ljungdahl pathway). In addition, glycolysis, the phosphate pathway, citrate cycle (TCA cycle or Krebs cycle), glyoxylate cycle, reductive Acetyl-CoA pathway, reducing pentose phosphate cycle, dicarboxyl ate-hydroxybutyrate cycle, and reducing citrate cycle were found in all MAGs. The carbohydrate-active enzyme (CAZyme) genes were examined in MAGs, and it was confirmed that all three strains (strain CJ11^T^, *L. tolerans* UM1^T^, and *L. silvestris* AM20-91^T^) lacked genes related to carbohydrate decomposition, such as xylan and chitin (Appendix A). Bacterial alcohol production is known to be commonly used in the production of alcoholic beverages and can be used as an important renewable energy source in the production of biofuels such as ethanol. However, this study does not suggest that strain CJ11^T^ fulfills the aforementioned roles.

### 3.3. Bacterial Growth and EPS Production Kinetics

Growth and fermentation kinetics are required for cultivating strain CJ11^T^ and determining the maximum EPS yield. When the novel strain CJ11^T^ was inoculated into a fresh sterile medium containing 1% (*w*/*v*) galactose, mannose, and glucose, it showed rapid cell growth for 24 to 48 h in a medium with mannose and galactose. The medium with glucose showed rapid cell growth between 48 and 60 h (Figure 5a). The logarithmic phase was confirmed between 48 and 72 h in the liquid medium supplemented with 1% glucose, and the stationary phase was shown for about 72 h after approximately 72 h. In addition, we elucidated the cell cycle of strain CJ11^T^ and identified the step-by-step fermentation kinetics of EPS production. The lyophilization of the EPS showed the highest yield in the medium supplemented with 1% glucose (Figure 5b). The bacterial growth curves over time for 0–7 days showed a classical pattern, and the EPS yields were generated faster in the medium supplemented with galactose and mannose within 96 h. However, after 96 h, the medium supplemented with glucose showed a higher yield than the medium supplemented with galactose and mannose, and at 144 h, the maximum yield was 1.2 g/L, which was confirmed to be higher than the EPS yield of the previously studied strains of the genus *Lysobacter* [13].

### 3.4. Chemical Analysis, Average Mw, and Monosaccharide Composition of EPS

The Lowry assay and bovine serum albumin (BSA) method of protein determination indicated that EPS contains 1.2% (*w*/*w*) protein content. The percentage of the carbohydrate content in lyophilized EPS was 83.8% (*w*/*w*). The protein content of the EPS produced by the novel strain, *Lysobacter soyae* CJ11^T^, was lower than that of *Lysobacter* sp. MMG2 [13]. As a result of measuring the average molecular weight of EPS produced by the new strain CJ11, it was confirmed to be 0.93 × 10^5^ Da (Appendix A). Although similar to the EPS and average molecular weight of previously published *Lysobacter* species, it was confirmed that the average molecular weight of the EPS produced by our strain was slightly lower [13]. Meanwhile, it was confirmed that the molecular weight was lower than that of *Bacillus haynesii* CamB6 [10]. Mw is a parameter that affects the functional properties of a exopolysaccharide and can vary depending on the nature of the starting material, the extraction temperature of the exopolysaccharide, and the fractionation method used [75,76].

The EPS was hydrolyzed with 2 M trifluoracetic acid, and monosaccharide composition was confirmed by Bio-LC analysis. In the acid-hydrolyzed EPS, four peaks were observed: mannose (29.53 min), glucose (24.04 min), galactose (20.93 min), and arabinose (19.56 min). Among them, mannose (72.2%) accounted for the overwhelming majority, followed by glucose (17.2%), galactose (7.8%), and arabinose (2.8%; Appendix A). Arabinose, on the other hand, is not a sugar commonly found in EPS produced by bacteria [77]. Previous studies have shown that arabinose partially inhibits sucrase activity and lowers the insulin peak [78]. Therefore, EPS containing arabinose may be useful. However, in this study, the characteristics of arabinose as an EPS composition of strain CJ11^T^ were not studied.

### 3.5. FE-SEM, TEM, and FTIR Analyses

A scanning electron microscope can identify the morphology and surface of the EPS microstructure, making it easier to understand the physical characteristics. The surface structure of the EPS produced by CJ11^T^ was observed at 10,000× and 30,000×. The surface morphology of the EPS, observed by SEM, was rough, irregular, and bumpy (Figure 6a,b). EPSs (white arrow) were observed on the surface of CJ11^T^ cells grown on R2A agar plates supplemented with 1% glucose. In contrast, the EPS layer was hardly observable on the surfaces of cells grown on R2A medium not supplemented with a carbon source, and the cells appeared rod-shaped (Figure 6c,d).

FTIR analysis was applied to confirm the presence of functional groups in EPS (Figure 7). A hydroxyl expansion vibration of the polysaccharide band was observed at 3281 cm^−1^ of the FTIR peak of the CJ11^T^ EPS [79], thus suggesting that this polymer is EPS. The peak observed around 2933 cm^−1^ in the spectrum corresponding to the methyl group was due to the C–H stretching vibration [80]. A 1631 cm^−1^ peak was found predicting the presence of a C=O group, which showed similar results in our previous study [81]. A peak corresponding to the N–H vibration of the amine group (peptide or proteins) was observed at 1540 cm^−1^ [82]. The peaks appearing around 1412 cm^−1^ and 1223 cm^−1^ were presumed to be COO^−^ vibrations and O–S–O groups, which are evidence of sulfuric acid esters, respectively [83]. The signal representing the stretching vibration of C–O and the changing angle vibration of O–H appeared around 1048 cm^−1^ [84]. The absorption band at 1048 cm^−1^, which is in the range of 1000–1200 cm^−1^, corresponded to the presence of carbohydrates [85]. The peak near 915 cm^−1^ was suggested to be pyranose [86]. FTIR analysis with other species of the same *Lysobacter* genus showed a similar profile [13].

### 3.6. XRD and TGA Analysis of EPS

As shown in Figure 8a, the XRD of the EPS was amorphous and showed a broad peak approximately from 5° to 20° (2θ). The XRD result is an indication that the EPS is in an amorphous state, and the EPS pattern shows a similar result to the EPS produced by *Lysobacter* sp. MMG2 [13]. The XRD pattern is a frequently used tool for confirming the crystallization and properties of materials, and crystallization analysis is important because it greatly reflects the thermal properties of materials and affects their temperature [87].

The thermal stability of EPS is important in industrial applications where sterilization is required, such as food processing and manufacturing [88]. Looking at the TGA analysis results in Figure 8b, the descending line (a) represents the weight loss of the EPS generated during the heating process, and the blue line “(b)” represents the heat flux (mW). The first mass loss occurred between 28 and 92.7 °C. This first weight loss was primarily due to gelatinization and swelling associated with water loss [89]. A weight loss of 8.08% at 210 °C indicated the onset of energy release associated with a maximum exothermic peak. At roughly 300 °C, the weight of the EPS dropped dramatically, with a mass loss of 29.9%.

### 3.7. ^1^H- and ^13^C-NMR Analysis

The structure of the EPS produced by *L. soyae* CJ11^T^ was established on the basis of one-dimensional NMR spectra (^13^C and ^1^H). The ^1^H-NMR spectrum of the EPS from *L*. *soyae* CJ11^T^ is shown in Figure 9a, displaying 14 resonances at δ 5.42, 5.41, 4.79, 3.99, 3.97, 3.95, 3.88, 3.59, 3.84, 3.82, 3.69, 3.67, 3.63, and 3.65 ppm. In the ^1^H-NMR result of the EPS produced by *L. soyae* CJ11^T^, one anomeric signal was detected at δ 4.79 ppm [90]. ^1^H-NMR is usually used to determine the glycosidic bond structure of polysaccharides. The EPS showed a signal lower than δ 4.8 ppm; therefore, it is a proton signal for *β*-anomer pyranose. FTIR analysis showed a band for pyranose; hence, the two results were the same [91]. The ^13^C-NMR spectrum of the EPS produced by *L. soyae* CJ11^T^ showed the signal of one anomeric region (δ_H_ 95–110 ppm) at δ_H_ 95.576 ppm [92]. Ring carbons were found at δ_H_ 50–85 ppm, alkyl carbons were found at δ_H_ 15–25 ppm, and carbonyl carbons were found at δ_H_ 165–180 ppm [92]. As shown in Figure 9b, on the basis of the chemical shift of the ^13^C-NMR, one signal was detected in the anomeric region, indicating the presence of that obtained from the ^1^H-NMR results.

### 3.8. DPPH Radical Scavenging of EPS

DPPH scavenging activity was observed in the EPS produced by *L. soyae*, which was lower than that of ascorbic acid used as a positive control (Figure 10). At a concentration of 0.25 mg/mL, EPS and ascorbic acid showed DPPH free-radical-scavenging activities of 71.85% and 95.36%, respectively, indicating that the scavenging activity of EPS was lower than that of ascorbic acid. As EPS increased to 4 mg/mL, DPPH free-radical-scavenging activity increased by 1.62% to 73.47%. This result showed higher activity at lower concentrations compared to the EPS isolated from *Lysobacter* sp. MMG2. These results indicate that the EPS produced by *L. soyae* influences free-radical scavenging. The data presented here indicate that strain CJ11^T^ has the potential to be utilized as a potential natural antioxidant; however, it was inferior to the positive control, ascorbic acid, in antioxidant capacity. In addition, DPPH is a commonly used compound to evaluate free-radical scavenging ability. EPSs are nontoxic and show high antioxidant capacity in vitro and in vivo, attracting attention as promising antioxidants [93]. The results were statistically analyzed using one-way ANOVA (analyzed using Prism GraphPad) and showed significance.

### 3.9. Description of Lysobacter soyae sp. nov

*Lysobacter soyae* (so’yae. N.L. gen. n. *soyae*, of soya, of soybean [referring to the source of the type strain]) cells are Gram-negative, aerobic, nonmotile, non-spore-forming, rod-shaped, 0.45–0.51 µm long, and 0.85–1.10 µm wide. Growth occurs on NA and LB agar, with R2A and TSA being the optimal medium. When grown on R2A agar plates at 30 °C for 3 days, colonies appear to be yellow, circular, and convex. Growth also occurs at 15–37 °C (optimum, 30 °C), pH 5.0–11.0 (optimum, pH 7.0–8.0), and 0–2% NaCl (optimum, 0%; *w*/*v*). Although the cells could not hydrolyze CM-cellulose, casein, chitin, Tween-80, and DNase, they hydrolyzed Tween-20. Its catalase activity was positive, whereas oxidase activity was negative. In the API 20NE tests, strain CJ11^T^ was positive for *β*-galactosidase and the assimilation of potassium gluconate, adipate, malate, and trisodium citrate. A negative reaction was observed for the nitrate reaction test and the fermentation of _D_-glucose, production of L-tryptophan, L-arginine, urease, and *β*-glucosidase, and hydrolysis of gelatin; the assimilation of _D_-glucose, L-arabinose, _D_-mannitol, _D_-mannose, *N*-acetyl-*β*-glucosamine, _D_-maltose, caprate, and phenyl acetate was also negative. The cells contained PE, DPG, and PG as major polar lipids and two unidentified phospholipids as minor polar lipids. Ubiquinone Q-8 was predominant in the cells. API ZYM strips gave negative results for lipase (C14), esterase (C4), cystine arylamidase, valine arylamidase, trypsin, *α*-chymotrypsin, *α*-galactosidase, *α*-fucosidase, *α*-mannosidase, *N*-acetyl-*β*-glucosaminidase, *β*-glucosidase, *α*-glucosidase, *β*-glucuronidase, and *β*-galactosidase, but positive results for the production of alkaline phosphatase, esterase lipase (C8), acid phosphatase, leucine arylamidase, and naphthol-AS-BI-phosphohydrolase. The type strain of *Lysobacter soyae* is CJ11^T^ (type strain CJ11^T^ = KACC 21716^T^ = NBRC 114478^T^), which was isolated from the roots of *Glycine max* L. at Dongguk university, Goyang-si, Republic of Korea. The G + C content of the genomic DNA is 59.2 mol.%.

The GenBank/EMBL/DDBJ/PIR accession numbers of the 16S rRNA gene sequences and the whole-genome sequences of *Lysobacter soyae* CJ11^T^ are MN915129 and CP080544, respectively.

## 4. Conclusions

In this study, EPSs were extracted, isolated, and purified from the fermentation broth of the novel strain *L*. *soyae* CJ11^T^ supplemented with glucose. EPS produced by a novel strain was composed of mannose, glucose, galactose, and arabinose, of which mannose was the most dominant, and the average molecular weight of EPS was 0.93 × 10^5^ Da. Through ^13^C-NMR spectral analysis at δ 95.576 ppm, we found a signal of one anomeric region. In addition, one anomeric region was also detected on the basis of ^1^H-NMR spectral analysis at δ 4.79 ppm. SEM showed a rough, irregular, and bumpy structured surface morphology. In addition, EPS was additionally characterized by FTIR and XRD. Functional characterization of EPS was also performed, showing DPPH radical-scavenging activity in a concentration-dependent manner, indicating potential as a natural antioxidant and high thermal stability. To our knowledge, no previous studies have reported the properties of EPS produced by *Lysobacter* isolated from *Glycine max*. Thus, this study demonstrates the theoretical knowledge and potential that the EPS produced by the critical strain CJ11^T^ could benefit several industries. On the basis of our results, we proposed that CJ11^T^ could be a novel *Lyobacter* species, which we named *Lysobacter soyae* sp. nov. and designated as CJ11^T^.

## Figures and Tables

**Figure 1 microorganisms-11-01900-f001:**
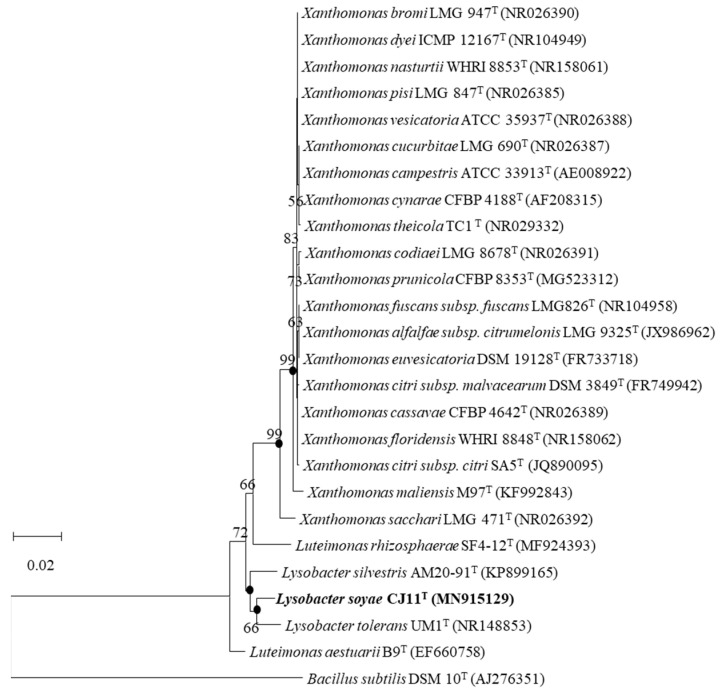
The NJ phylogenetic tree based on 16S rRNA gene sequences showing the position of strain CJ11^T^. Bootstrap values are shown as percentages of 1000 replicates (above 50%). Filled circles indicate that the corresponding nodes were recovered in trees generated using the maximum-parsimony and maximum-likelihood algorithms. GenBank accession numbers of 16S rRNA sequences are given in parentheses. *Bacillus subtilis* DSM 10^T^ (AJ276351) was used as the outgroup. Bar, 0.02 changes per position.

**Figure 2 microorganisms-11-01900-f002:**
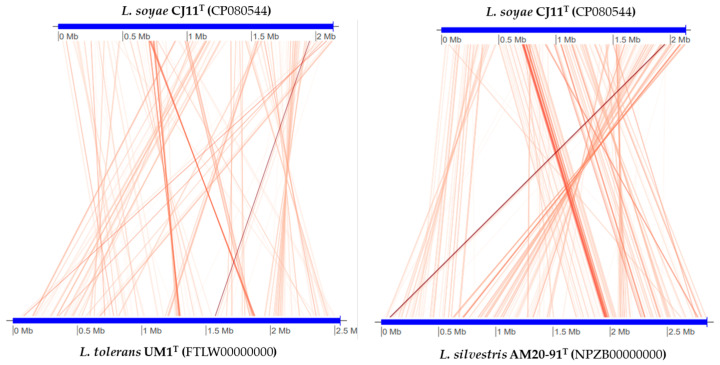
Illustration representing FastANI’s workflow between the novel strain CJ11^T^ genome and a phylogenetically close reference genome.

**Figure 3 microorganisms-11-01900-f003:**
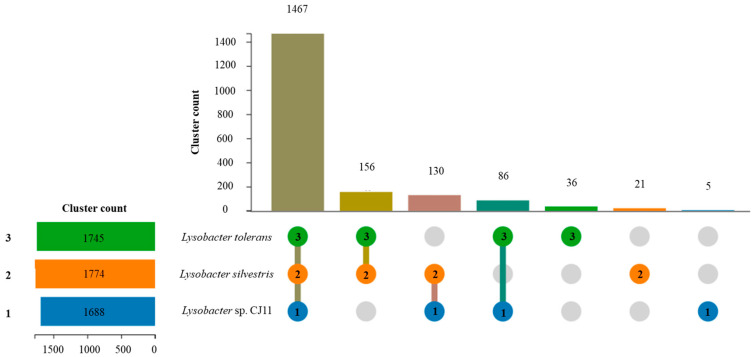
Using OrthoVenn3 showed the proteome comparison among the selected three *Lysobacter* species: (1) strain CJ11^T^ (blue); (2) *L. tolerans* UM1^T^ (green); (3) *L. silvestris* AM20-91^T^ (orange). There were 1467 orthologous gene clusters shared by three strains. When strain CJ11^T^ was compared with phylogenetically close *Lysobacter* species, 130 and 86 clusters were shared between strains 2 and 3, respectively. Strains 1, 2, and 3 had five, 21, and 36 unique clusters, respectively.

**Figure 4 microorganisms-11-01900-f004:**
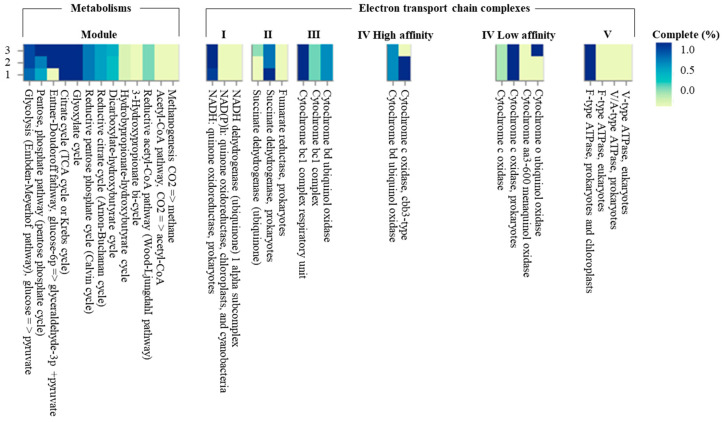
The heatmap constructed using Distilled and Refined Annotation of Metabolism (DRAM) shows a list of metabolites of important microbial traits in three *Lysobacter* species. (1) strain CJ11^T^; (2) *L. tolerans* UM1^T^; (3) *L. silvestris* AM20-91^T^.

**Figure 5 microorganisms-11-01900-f005:**
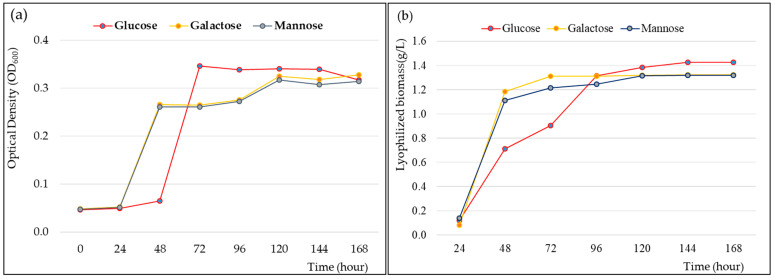
Cell growth rate (**a**) and exopolysaccharide (EPS) production (**b**) of *Lysobacter soyae* CJ11^T^ over time. Glucose, galactose, and mannose were used, respectively, as carbon sources.

**Figure 6 microorganisms-11-01900-f006:**
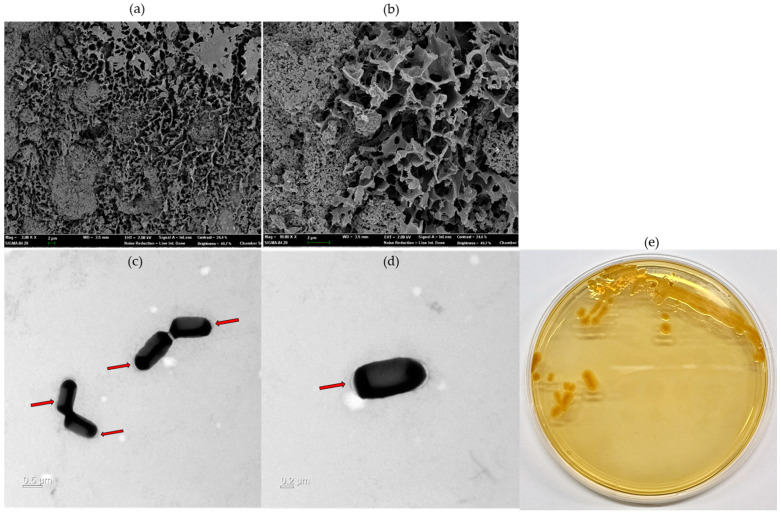
Microstructure image of exopolysaccharides (EPS) fabricated on *L*. *soyae* CJ11^T^ taken by SEM ((**a**), 10,000×; (**b**), 30,000×). Transmission electron microscope (TEM) images of *L*. *soyae* CJ11^T^ cells cultured in Reasoner’s 2A (R2A) broth supplemented with 1% glucose (*w*/*v*) for 72 h: (**c**) four cells; (**d**) single cell. Red arrows indicate EPS produced by *L. soyae* CJ11^T^ (scale bar = (**c**), 0.5 µm; (**d**), 0.5 µm). (**e**) Colony morphology of strain CJ11^T^.

**Figure 7 microorganisms-11-01900-f007:**
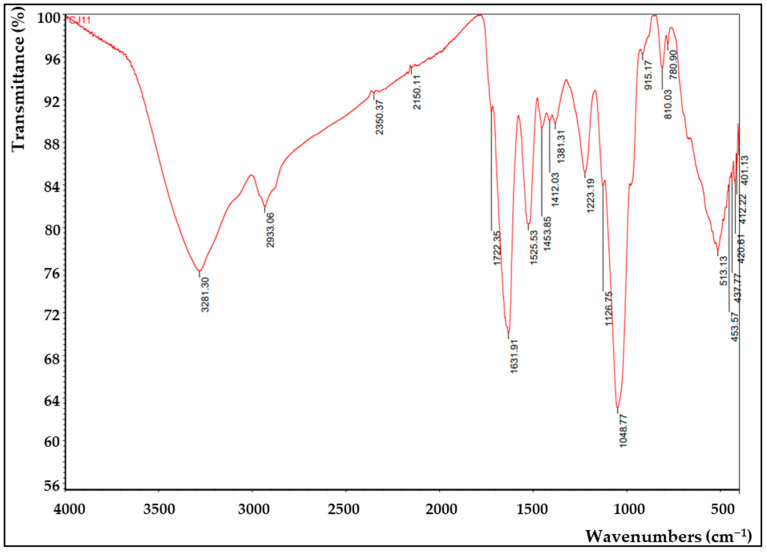
Fourier-transform infrared (FTIR) analysis of the EPS produced by *L. soyae* CJ11^T^.

**Figure 8 microorganisms-11-01900-f008:**
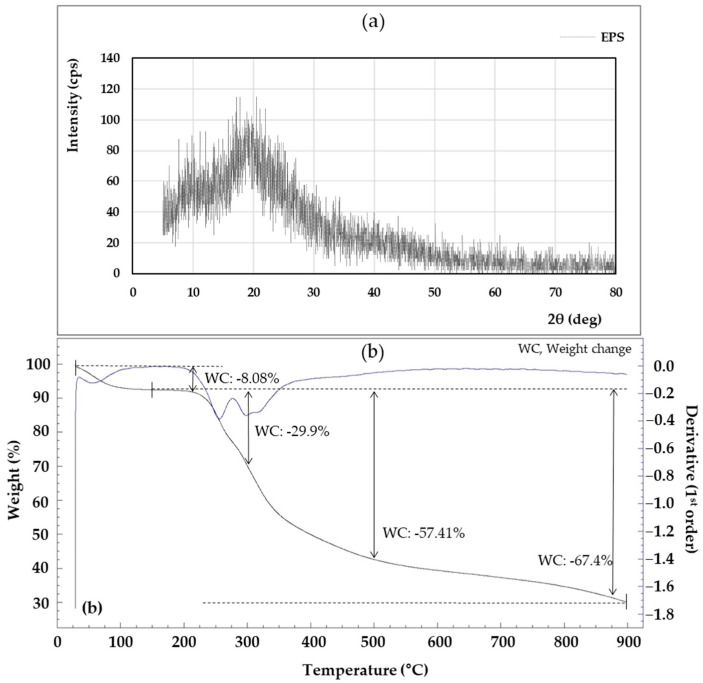
X-ray diffraction (XRD) (**a**) and thermogravimetric (TGA) (**b**) of the exopolysaccharides (EPS) produced by *L*. *soyae* CJ11^T^.

**Figure 9 microorganisms-11-01900-f009:**
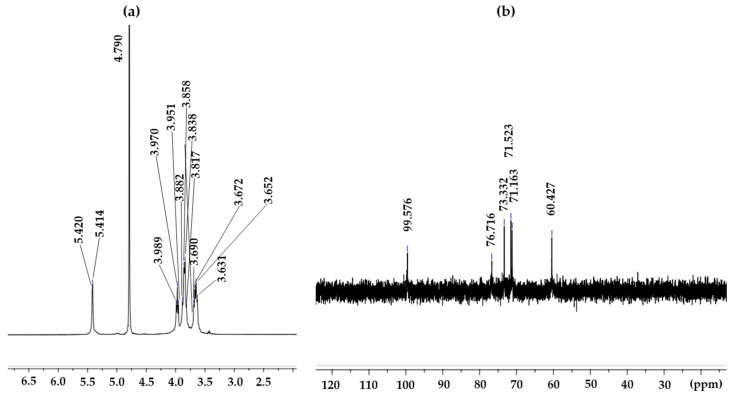
^1^H-NMR (**a**) and ^13^C-NMR (**b**) spectra of the *L*. *soyae* CJ11^T^ EPS.

**Figure 10 microorganisms-11-01900-f010:**
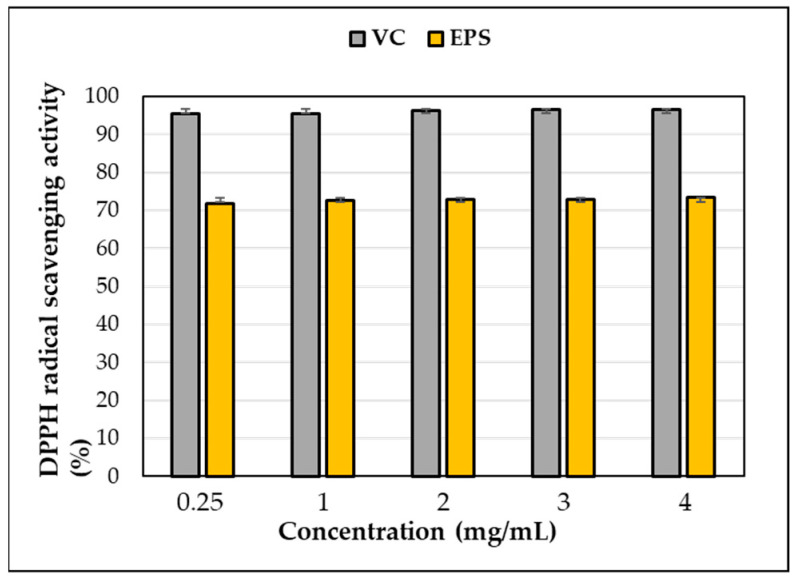
DPPH free-radical-scavenging activity of exopolysaccharides (EPS) produced by strain CJ11^T^. Error bars represent the standard deviation of the mean (n = 3).

## Data Availability

The GenBank/EMBL/DDBJ/PIR accession number for the 16S rRNA gene sequences of strain CJ11^T^ is MN915129. GenBank/EMBL/DDBJ/PIR accession number for the whole-genome sequence of strain CJ11^T^ is CP080544. The data that support the findings of this study are available from the corresponding author upon reasonable request.

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
