# Peer review of "Characterization and Antioxidant Activity of Exopolysaccharides Produced by Lysobacter soyae sp. nov Isolated from the Root of Glycine max L."

_microorganisms, 2023, doi:10.3390/microorganisms11081900_

Round 1
Reviewer 1 Report
In this manuscript, the authors screened a Lysobacter strain that can produce a new exopolysaccharide. This exopolysaccharide was subsequently purified and characterized. Overall, this manuscript is not yet in a publishable state. It needs some revisions. Below are detailed the most relevant:
1) Please explain the relationship between biopolymers and EPS. Line 33-34
2) “Lysobacterales” should be in italics. Line 40
3) This manuscript mentions that "the EPS generated from one Lysobacter species inhibited biofilm formation" in line 54, please explain whether EPS has this effect in this manuscript.
4) The layout of Figure 1 seems to be faulty. Please correct it.
5) It is recommended that micro and macro photographs of CJ11T should be included as supplementary materials.
6) Whether to indent the first line of line 387?
7) Does title 3.5 require bold font and first line indent? Line 400
8) Please provide Bio-LC peak diagram of monosaccharide standard in the supplement
9) Please analyze the data in Figure 10 for significance between groups, not within groups and there seems to be something wrong with the error rod data
10) The discussion section is weak, please improve it.
11) Exopolysaccharides are not protein-removed and purified by ion exchange resins, and the data from unpurified polysaccharides are not reliable.
12) Please provide molecular weight map.

Reviewer 2 Report
The manuscript entitled «Characterization and antioxidant activity of exopolysaccharides produced by Lysobacter soyae sp. nov isolated from the root of Glycine max L.” is devoted to
O-polysaccharide with molecular weight 0.93 × 105 Da isolation and characterisation from strain Lysobacter sp. CJ11T isolated from Glycine max L rhizosphere in Goyang-si, Republic of Korea. In this study, a significant amount of work has been carried out, including the isolation and characterization of a new strain from the rhizosphere, description its physiology and morphological characteristics, the isolation of the polysaccharide, its identification by NMR, FTIR, bio-liquid chromatography, gel permeation chromatography (GPC), and the assessment of its biochemical properties. Therefore, I believe that the manuscript aimed to an important and relevant topic and in general corresponding to the aims and scopes of the Microorganisms journal.
I have some remarks to the manuscript
1. I didn't like the introduction. The first paragraph is too general. It is necessary to answer the question why the strain from the rhizosphere was chosen, why the polysaccharides of strains from the rhizosphere can be interesting. The second paragraph has very detailed Lysobacter description. However, in general, the purpose of carrying out such a significant amount of work is not clear.
2. section 3.3. The genome features of strain CJ11T contains a lot of redundant information; however, since the work is devoted to the polysaccharide, the main attention should be paid to the description of the polysaccharide synthesis cluster.
3. Figure 2. (a), unreadable. It can be transferred to saplimentari and increase the font
4. what does the proteome comparison among the selected three Lysobacter species give for the general idea of the article? redundant information in my opinion
5. Figure 6. Move to supplementary
6. what does electron microscopy provide? Completely redundant information.
7. 1H and 13C NMR analysis is commonly used to establish the structure of the polysaccharide monomer. In this case, I have redundant information.
8. in Figure 10, the size of all columns is almost the same. Maybe it should not be given, but simply described in the text?
9. Section 3.8. instead of discussion? With such a quantity of material, a manuscript without discussion is absolutely impossible. It is necessary to comprehend all the points in a single concept. So far it looks like scattered facts, but there is no general material.
10. Conclusions do not appear to be conclusions because there is no general idea. As long as it looks like this. We have a lot of good equipment, a lot of good instrument operators, we have the task of using as many instruments as possible, so we isolated the strain and studied it with all the methods available.
11. in general, I advise the authors to divide the material into two articles, on the isolation and identification of a strain with physiological characteristics, and the second one, dedicated to the isolation of the identification and study of the properties of the polysaccharide
Minor editing of English language required
Round 2
Reviewer 1 Report
1. Maybe there is a problem with my presentation, in the previous suggestion "It is recommended to add micro and macro photos of CJ11T as supplementary material." Please refer to this article (https://doi.org/10.3390/molecules27217209) Fig. S2.
2. I suggest as in Fig. S8 should be incorporated in the manuscript.
3. it is recommended that the proportions of each monosaccharide component be calculated.
Reviewer 2 Report
In my opinion, the authors have significantly improved the quality of the manuscript and I can recommend it for publication in this form
Author Response
We didn't get any additional comments from reviewer 2.